# A portable regulatory RNA array design enables tunable and complex regulation across diverse bacteria

Baiyang Liu[1], Christian Cuba Samaniego[2], Matthew R. Bennett [3,4], Elisa Franco [2] & James Chappell [3,4] ✉

A lack of composable and tunable gene regulators has hindered efforts to engineer non-model bacteria and consortia. Toward addressing this, we explore the broad-host potential of small transcription activating RNA (STAR) and propose a design strategy to achieve tunable gene control. First, we demonstrate that STARs optimized for *E. coli* function across different Gram-negative species and can actuate using phage RNA polymerase, suggesting that RNA systems acting at the level of transcription are portable. Second, we explore an RNA design strategy that uses arrays of tandem and transcriptionally fused RNA regulators to precisely alter regulator concentration from 1 to 8 copies. This provides a simple means to predictably tune output gain across species and does not require access to large regulatory part libraries. Finally, we show RNA arrays can be used to achieve tunable cascading and multiplexing circuits across species, analogous to the motifs used in artificial neural networks.

Controlling gene expression is at the heart of how cells control function and phenotype[1,2]. Motivated by this, there has been a significant emphasis placed on creating libraries of DNA, RNA, and protein-based regulatory systems that enable the precise and advanced programming of gene expression[3,4]. As access to better genetic parts has been achieved, this has led to a dramatic acceleration in our ability to precisely control the expression of individual genes[5,6], optimize the expression of multi-gene pathways[7,8], and compose larger genetic circuits able to implement signal processing[9,10] and feedback control[11,12]. In bacteria, robust genetic programming frameworks exist for model microbes, most notably domesticated laboratory strains of *Escherichia coli*. However, our ability to program other diverse bacteria and non-model species is currently lacking. This is problematic because it fundamentally limits our ability to utilize diverse chassis that are naturally adapted for different environments (e.g., aquatic, soil, built environment, host-associated)[13], have distinct metabolic capabilities[14], or offer valuable phenotypes (e.g., sporulation, motility, electrogenesis)[15]. This also hinders efforts to program native microbial consortia and microbiomes that are predominantly composed of non-model and currently genetically intractable species. Thus, there is strong motivation to create genetic engineering frameworks for the programming of non-model species.

RNA regulators offer the intriguing potential of broad-host genetic control elements that could be functionally portable across diverse microbes. This is because most RNA switches rely upon the formation of simple structures (e.g., duplexes, hairpins) that depend upon universal RNA interactions. Additionally, most RNA switches use RNA motifs (e.g., RBS, transcriptional terminators) that have relatively conserved sequence-function relations[16–18], and enact regulation through interactions with host-cell machinery that is ubiquitous. While there have been a handful of demonstrations of the portability of synthetic RNA switches[19–23], we currently lack systematic investigation. Additionally, it remains hard to tune input-output relations (i.e., gain). For example, while RNA engineering can be applied to alter gain[6,24–30], this requires a detailed understanding of the system and is often arduous. A more convenient approach is to tune the expression of the

[1]Graduate Program in Systems, Synthetic, and Physical Biology, Rice University, Houston, TX, USA. [2]Department of Mechanical and Aerospace Engineering, Bioengineering, Molecular Biology Institute, University of California at Los Angeles, Los Angeles, CA, USA. [3]Department of Biosciences, Rice University, Houston, TX, USA. [4]Department of Bioengineering, Rice University, Houston, TX, USA. ✉e-mail: jc125@rice.edu

RNA components of a given switch through the use of promoter strength libraries[6,29,31]. However, this can often be unpredictable due to non-linear transfer functions of RNA switches and requires access to large promoter libraries that are often lacking in microbes beyond *E. coli*. Thus, while RNA offers an intriguing solution for engineering diverse bacterial species, there is a need to investigate portability and create simple strategies to tune these genetic parts across different hosts.

In the present study, we verify the portability of a synthetic RNA switch called small transcription activating RNA (STAR) across different Gram-negative species and cellular contexts. We then establish a design strategy to tune STAR output gain. This strategy that we call regulatory RNA arrays is inspired by naturally occurring CRISPR arrays that convert a single transcript into separate and independently acting RNA species. Using this approach, we show we can precisely alter regulator concentration from 1 to 8 copies, which in turn, results in corresponding changes in output gain. This approach works across different species and can achieve tunable gene control without the need for auxiliary part libraries. Finally, we show RNA arrays can be used across species to achieve modular signaling cascades with minimal signal loss, and to process a single input into multiple outputs achieving a tunable weighting comparable to a neural network motif.

## Results

### STAR regulators allow for robust activation in diverse cellular contexts

We first sought to understand the portability of RNA switches across different species and cellular contexts. To investigate this, we choose a synthetic RNA switch called STAR (Fig. 1A)[26,29]. This switch is composed of a target RNA placed upstream of a gene to be controlled, which by default folds into a terminator hairpin turning the downstream gene off. This switch can be activated by transcribing a STAR, which prevents terminator formation and turns on the transcription of the gene. As this switch functions at the level of transcription—converting an RNA input into an RNA output—it is only dependent upon RNA polymerase (RNAP) and its ability to initiate and terminate transcription. Changing RNAP by implementing the switch in different hosts or using non-native RNAP has the potential to introduce a disturbance of the input-output relationship; however, we hypothesize that this disturbance will be minimal given the ubiquitous nature of RNA terminator motifs (Fig. 1B). To investigate this disturbance, we chose a panel of different Gram-negative bacteria derived from diverse environments of interest to synthetic biologists (e.g., gut, soils, aquatic, marine). Specifically, we chose *E. coli*, *Shewanella oneidensis*, *Pseudomonas fluorescens*, *Pseudomonas putida*, *Pseudomonas stutzeri*, and *Vibrio*

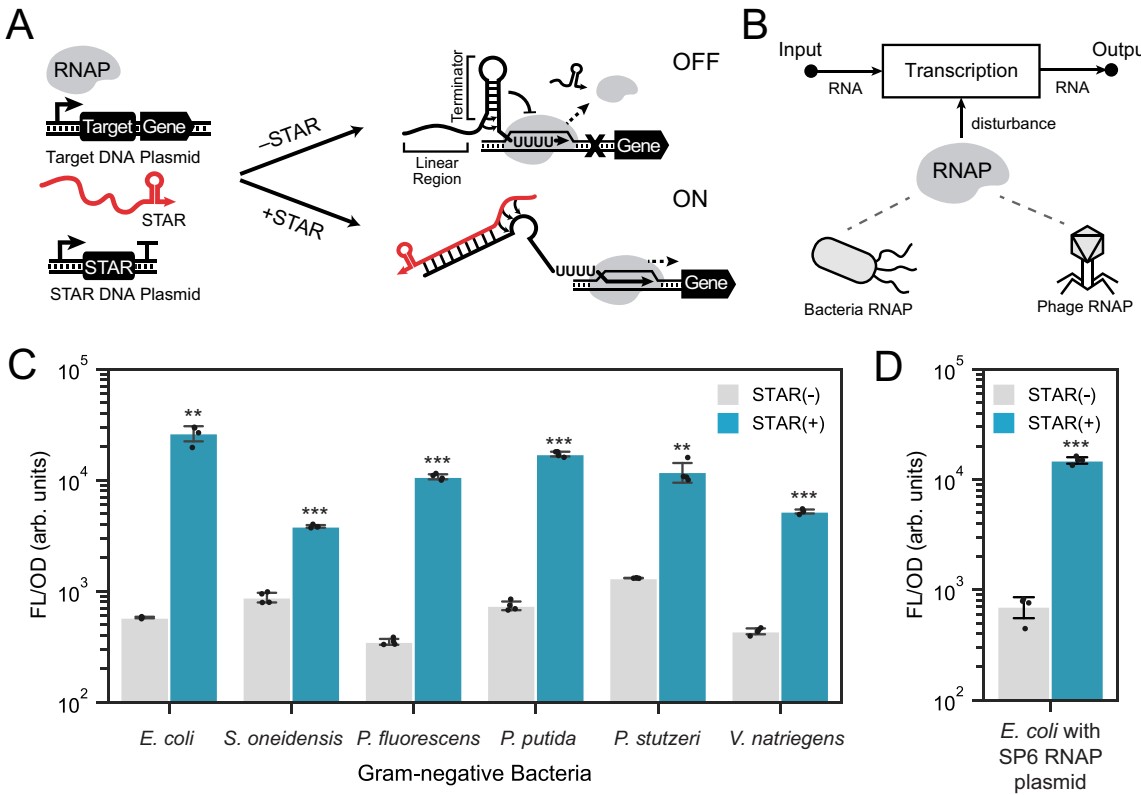

**Fig. 1 | STAR regulators can function with different Gram-negative bacterial and phage RNA polymerase. A** STAR mechanism. STAR regulators are composed of a target RNA containing a terminator hairpin placed upstream of an output gene and a complementary small transcription activating RNA (STAR). A constitutive promoter initiates transcription of the target RNA, which by default, folds into a terminator to stop the transcription by RNA polymerase (RNAP), turning the gene off. In the presence of the STAR, the target RNA folds into an alternative anti-terminated structure, which allows transcription of the output RNA. **B** Schematic of STAR regulators that act transcriptionally with RNA serving as both the input and output. The main cellular factor required is RNAP which if changed (e.g., different species or with phage-derived RNAP) has the potential to disturb the input-output relationship. **C** STAR characterization in diverse Gram-negative bacteria. Fluorescent characterization in cells transformed with the same set of broad-host range

plasmids containing target RNA-controlled GFP in the absence [STAR(-)] and presence [STAR(+)] of a STAR-encoding cassette. The fold activation from left to right are 45.7, 4.4, 30.6, 23.2, 9.0, and 12.0. The *p* values for the t-test from left to right are 1.66e-03, 3.52e-08, 6.73e-05, 5.36e-05, 4.64e-03, 3.13e-05. **D** STAR characterization with phage-derived RNAP. Fluorescent characterization in *E. coli* cells transformed with a plasmid containing target RNA-controlled GFP in the absence [STAR(-)] and presence [STAR(+)] of a STAR-encoding cassette. SP6 phage RNAP promoters are used to express STAR and target RNA, and SP6 RNAP is expressed heterologously. The fold activation is 21.2. The *p*-value for the t-test is 1.13e-04. Bars show mean values and error bars represent s.d. of *n* = 4 biological replicates shown as points. Two-tailed t-tests assuming unequal variance were used (*df* = 6), and the significance is marked by asterisks above bars indicating $p < .05$ (*), $p < .01$ (**), $p < .001$ (***). Source data are provided as a Source Data file.

*natriegens*. Each species was transformed with the same set of broad-range plasmids encoding a target RNA-controlled GFP along with either a STAR-expressing cassette or without this cassette. Cell fluorescence measurements were then performed on these transformants (Fig. 1C). From these experiments, we observed robust activation across different species, with a statistically significant increase in fluorescence observed in the presence of STAR. Importantly, this activation was achieved without any modification required to the RNA switch sequence or the expression system (e.g., promoters and plasmid backbones).

Motivated by these observations, we next investigated if STARs would work with the SP6 phage RNAP, which is known to function in diverse contexts[32]. This single-subunit RNAP is distinct from the multi-subunit RNAP present in bacteria in their structure, promoter recognition, and enzymatic activity[33,34]. To test if STARs were compatible with SP6 RNAP, we replaced the promoters driving the target RNA and STAR with SP6-specific promoters and tested STAR activity in *E. coli* cells expressing SP6 RNAP (Fig. 1D). Surprisingly, we observed STARs were functional and able to efficiently terminate transcription in the absence of STAR and activate in its presence. Given the differences in the kinetics of phage-derived RNAP, having ~6-fold higher elongation rates than bacterial RNAP[35–37], we next sought to understand if this impacted STAR design rules. Interestingly, as shown for STARs operating with *E. coli* RNAP[29], STARs operating with an SP6 RNAP have an optimal length of STAR linear region of ~40 nt, could regulate STARs orthogonally, and have an output that is affected by STAR concentration (Supplementary Fig. 1). Taken together, these results show that STARs can function robustly across different Gram-negative species and using phage RNAP with minimal modification. This confirms one of our central hypotheses, that RNA-only genetic parts can function across different cellular contexts with minimal disturbance.

## Creating a regulatory RNA array design strategy for predictable tuning of output gain

We next sought to create a simple strategy to tune the input-output relations of STAR switches. As increasing the cellular concentration of STAR is known to increase the output signal[29], we reasoned that producing multiple copies of STAR from a single promoter input would provide an effective, predictable, and simple approach to amplify the output gain (Fig. 2A). Inspired by the natural organization of CRISPR arrays, we expected that synthetic arrays of regulatory RNA could be created that achieved our design goals. Specifically, we envisaged using RNA cleavage sites to insulate and separate tandem STARs from a single transcript, producing separate molecular species that can function independently. As this strategy converts a single transcriptional event into multiple RNA outputs, we hypothesized it would serve as a tunable genetic amplifier[38]. To investigate this, we first aimed to identify an effective RNA insulator sequence that could be used to separate and insulate tandem STARs. Given their widespread use in genetic circuitry, we first tested a self-cleaving ribozyme PlmJ. To do this, we compared the transcription activation from a construct encoding a single STAR insulated by PlmJ (x1) and a construct encoding four tandem STARs separated by PlmJ (x4). For these experiments, we focused on STAR variant 50 which is from a previously described orthogonal library[29]. We posited that effective insulation would result in efficient separation of tandem STARs, allowing each to act independently and increase the level of activation proportionally to STAR copy. To allow for the efficient cloning of RNA arrays, we applied a modular cloning approach based on Golden Gate cloning (Supplementary Fig. 2). However, our initial experiment indicated that tandem STARs performed poorly when using PlmJ, resulting in little increase in activation from four copies compared to a single STAR (Fig. 2B) (Supplementary Fig. 3). Computational structure prediction analysis suggested that this might arise because of misfolding of the transcript that led to an RNA structure that could neither be cleaved or activate

transcription (Fig. 2B) (Supplementary Fig. 4). To address this, we investigated the use of the CRISPR-associated endonuclease Csy4 and hairpin (csy4hp), which has naturally evolved to separate often repetitive tandem CRISPR arrays and has been used previously as an insulator[39]. In addition, as computational structure prediction analysis suggested that the use of csy4hp alone could result in a partially misfolded transcript, we also investigated a design in which csy4hp was preceded by a strong hairpin structure (shcsy4hp), which structure prediction suggested would fold more robustly (Fig. 2B) (Supplementary figure 3). As an additional validation, we also confirmed that a single STAR maintained similar performance in the presence and absence of the csy4hp or shcsy4hp insulators, suggesting robust folding (Supplementary Fig. 5). To test these insulators in regulatory RNA arrays, a single plasmid design encoding both the target RNA-controlled GFP and the STAR arrays was constructed and co-transformed into cells alongside a constitutively expressed Csy4 plasmid. Testing the csy4hp and shcsy4hp designs, we observed proportional activation from increased copies of STARs, suggesting we are achieving effective insulation and that predictable activation could be achieved by tuning STAR copy number.

Next, we aimed to confirm the approach could be applied to other STAR variants with distinct sequences and characteristics. To do this, we tested the shcsy4hp design on STAR variant 10, which is another STAR from the orthogonal library[29]. We observed comparable results to those obtained using STAR variant 50, in which increasing activation was observed with increasing STAR copy. Taken together, this data provided the basis for creating regulatory RNA arrays and suggested that controlling STAR copy would provide a simple approach for predictably tuning output gain.

## Regulatory RNA arrays provide a portable approach to predictably tune output gain

With an effective insulation strategy for regulatory RNA arrays in hand, we next investigated the precise relation between STAR copy and output signal, and the portability of this approach across different microbes (Fig. 3A). First, using the shcsy4hp as an insulator we created constructs encoding 1, 2, 4, 6, and 8 STAR copies in a single transcript using a custom modular cloning (MoClo) approach (Supplementary Fig. 2). As our ultimate goal was to test this strategy across multiple hosts, these RNA arrays were encoded on a broad-host plasmid, pBBR1, and downstream of the arabinose (pBAD) or cumate inducible promoter (pCymR) that were confirmed to function in *S. oneidensis* and *P. fluorescens* respectively in preliminary testing. A target RNA-controlled GFP downstream of a constitutive promoter was cloned onto the same plasmid. We first performed fluorescence characterization of the different arrays using pBAD and pCymR in *E. coli* cells. As the relation between inducer concentration and the relative promoter activity (i.e., transcription output) of different inducible systems is often non-linear, we converted the inducer concentrations into a relative promoter activity term. To do this, we measured the transcription output of each inducible promoter system using a GFP reporter under the same experimental conditions to create a series of standard curves (Supplementary Fig. 6). This was used to convert the inducer concentration used in the array characterization into an expected input promoter activity (Supplementary Fig. 7). From these data we observed a strong correlation between STAR copy and transcription output in *E. coli* cells (Fig. 3B, C) (Pearson correlation coefficient to a straight-line model of $r = 0.954$ and $0.955$). This linear relation between array copy and output was observed across a range of different induction levels (Supplementary Fig. 8). Interestingly, the STAR copy did not appear to significantly alter the off-state expression in the absence of inducer (i.e., leakiness) (Supplementary Fig. 9).

We next characterized the performance of these plasmids in *S. oneidensis* and in *P. fluorescens* using arrays controlled by pBAD and pCymR respectively. We observed proportional relation

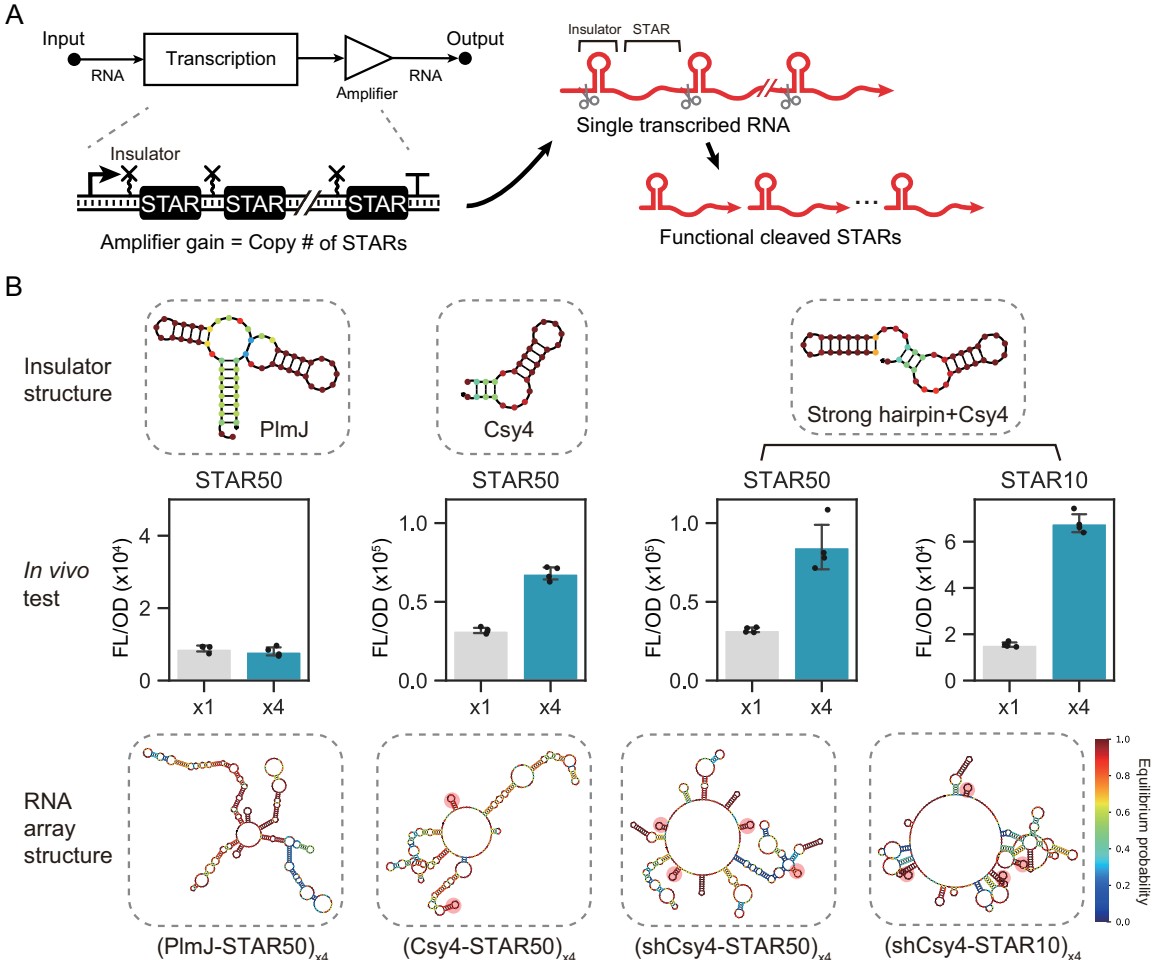

**Fig. 2 | Investigating insulation strategies for regulatory RNA arrays.**
**A** Schematic of regulatory RNA arrays. Tandem repeats of STARs are used to convert a single RNA input into multiple RNA outputs to achieve amplification of gain. Insulators cleave the arrays to allow physical decoupling and production of individual output RNA. **B** Structure predictions and in vivo characterization of different RNA-based insulators. Top panel shows structure predictions of the insulators: PlmJ hammerhead ribozyme, csy4 ribonuclease cleavage hairpin (csy4hp), and a strong hairpin-csy4 fusion (shcsy4hp). Middle panel shows fluorescent characterization in *E. coli* cells transformed with a target RNA-controlled

GFP in the presence of regulatory RNA arrays containing a single (x1) or four STAR (x4) copies. Bars show mean values and error bars represent s.d. of $n = 4$ biological replicates shown as points. Two-tailed t-tests assuming unequal variance were used ($df = 6$), and significance is marked by asterisks above bars indicating $p < .05$ (*), $p < .01$ (**), $p < .001$ (***). The $p$ values for the t-test from left to right are 0.386, 7.06e-06, 3.88e-06, 3.16e-05. Bottom panel shows the structure prediction of the regulatory RNA arrays containing 4 tandem STAR repeats of STAR variant 10 (STAR10) and 50 (STAR50) separated by each insulator (shaded in red if the insulator is folded correctly). Source data are provided as a Source Data file.

between transcription output and STAR copy across a range of induction levels (Fig. 3D, E, and Supplementary Fig. 8). As before, STAR copy did not significantly alter off-state expression in absence of the inducers (Supplementary Fig. 9). Taken together, these results confirm that regulatory RNA arrays provide a simple, predictable, and portable approach to amplify and tune output signals across diverse microbes.

One concern of using multiple repeats of RNA sequences in the regulatory RNA array design is the risk of genetic instability due to recombination. To understand if this was a problem, we tested the stability of the plasmids in *E. coli* cells by sequencing plasmids across a 5-day consecutive culture. Surprisingly, we observed the plasmids appear to be stable even with 8 repeats of STAR after 5 days of culturing (Supplementary Fig. 10).

We next constructed a simple mathematical model to resolve a deeper understanding of regulatory RNA arrays and our experimental results (Supplementary Note 1). This model consists of two parts: the production and cleavage of regulatory RNA arrays through Csy4 and

the activation of the target RNA by STAR, which builds upon a prior model[38]. In our data, we observed a linear relationship between STAR copy and output gain across diverse cell contexts. To understand the robustness of this linear relation, we used our model to investigate the effect of different cleavage efficiencies and RNA array transcription rates (Fig. 3F). While reduced cleavage efficiencies show a decreased output gain, this did not affect the linear relation. On the other hand, RNA array transcription input can induce saturation of the output gain at high levels (Fig. 3F, Supplementary Fig. 11). In most cases we do not observe saturation in our experimental data, suggesting we are below this threshold. The one exception to this is in *E. coli* using a pBAD input (Fig. 3B), which appears to begin to saturate at high induction levels. Thus, our model suggests that RNA arrays should robustly maintain a linear relation between STAR copy and output gain, as long as the target RNA is not saturated.

Another feature of our data we wanted to explore was the effect of leaky expression from the target RNA on the off state. Adjusting this parameter in our model, we observed that leakiness does not

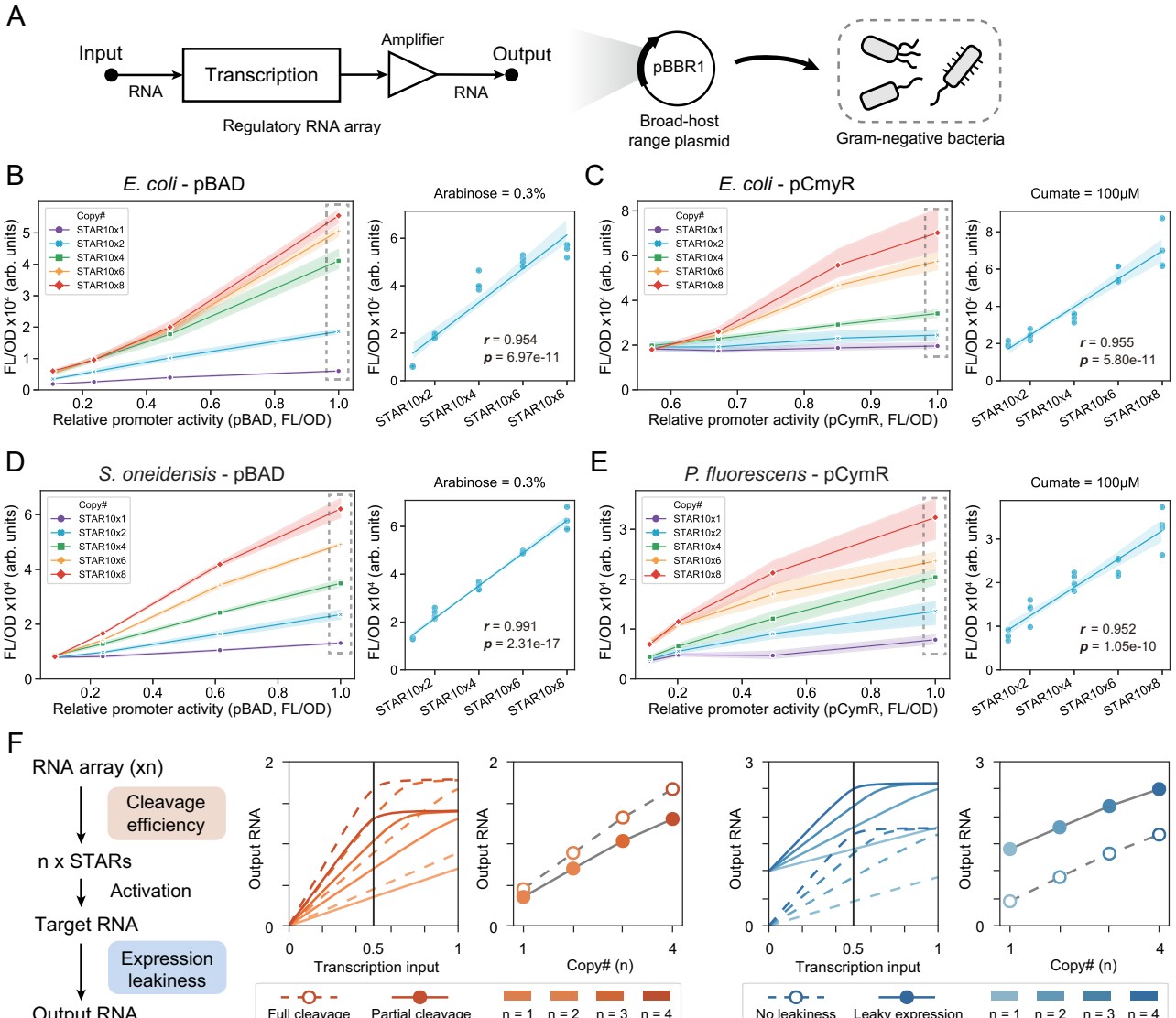

**Fig. 3 | Regulatory RNA arrays function predictably and are portable across diverse species. A** Schematic of broad-host regulatory RNA arrays. RNA arrays can be cloned onto a broad-host plasmid to allow implementation in diverse species. **B** Arabinose-inducible regulatory RNA arrays and (**C**) cumate-inducible regulatory RNA arrays in *E. coli*. (**D**) Arabinose-inducible regulatory RNA arrays in *S. oneidensis* and (**E**) cumate-inducible regulatory RNA arrays in *P. fluorescens*. In (**B**)−(**E**), left panels show fluorescent characterization in cells transformed with a plasmid containing a target RNA-controlled GFP and an inducible STAR array encoding 1, 2, 4, 6, and 8 copies. The relative promoter activity for each inducer condition is calculated based on data from a corresponding inducible GFP control. Each point shows the mean value, and the shade represents s.d. of $n = 4$ biological replicates. The right panels show the correlation between the STAR copy number in the RNA array and the corresponding fluorescent output at the promoter activity indicated by the dashed box in the left panel. Pearson correlation coefficient and corresponding p-value are calculated ($df = 18$) and shown as *r* and *p*. **F** (Left panel)

Schematic of the model to investigate the effect of cleaving efficiency and leaky transcription in the off state. (Center panel) Simulation data of the input-output behavior between the rate of transcription input into the RNA array and the concentration of the output RNA for multiple STAR copies (increase from light orange to red). Units for transcription input and output RNA are arbitrary units. The dashed line (with empty circles) and the solid line (with solid circles) represent high and low cleaving efficiencies. For a constant transcription input of 0.5 (shown in a vertical black line), we plot the relation between STAR copies and RNA output. (Right panel) The effect of leakiness on the input-output between transcription input and RNA output. The dashed line (with empty circles) and the solid line (with solid circle) represent low and high off state leaky transcription. For a constant transcription input of 0.5 (shown in a vertical black line), we plot the relation between STAR copies and RNA output. Source data are provided as a Source Data file.

affect the linear relation between STAR copy number and output gain. However, leakiness from the target RNA causes an upward shift of the curves, regardless of STAR copy number (Fig. 3F). Overall, our simple model can capture the linear relation between STAR copy number and output gain that we observed in our data across diverse cellular contexts and suggests a level of robustness to RNA array performance.

## Regulatory RNA arrays are composable and tunable for broad-host genetic circuitry

The construction of synthetic genetic circuits analogous to those seen in natural systems has been a long-standing goal of synthetic biology. We next wanted to demonstrate that regulatory RNA arrays could be utilized to advance the creation of tunable RNA-only genetic circuits for non-model species[26,29,40,41]. To begin with, we wanted to

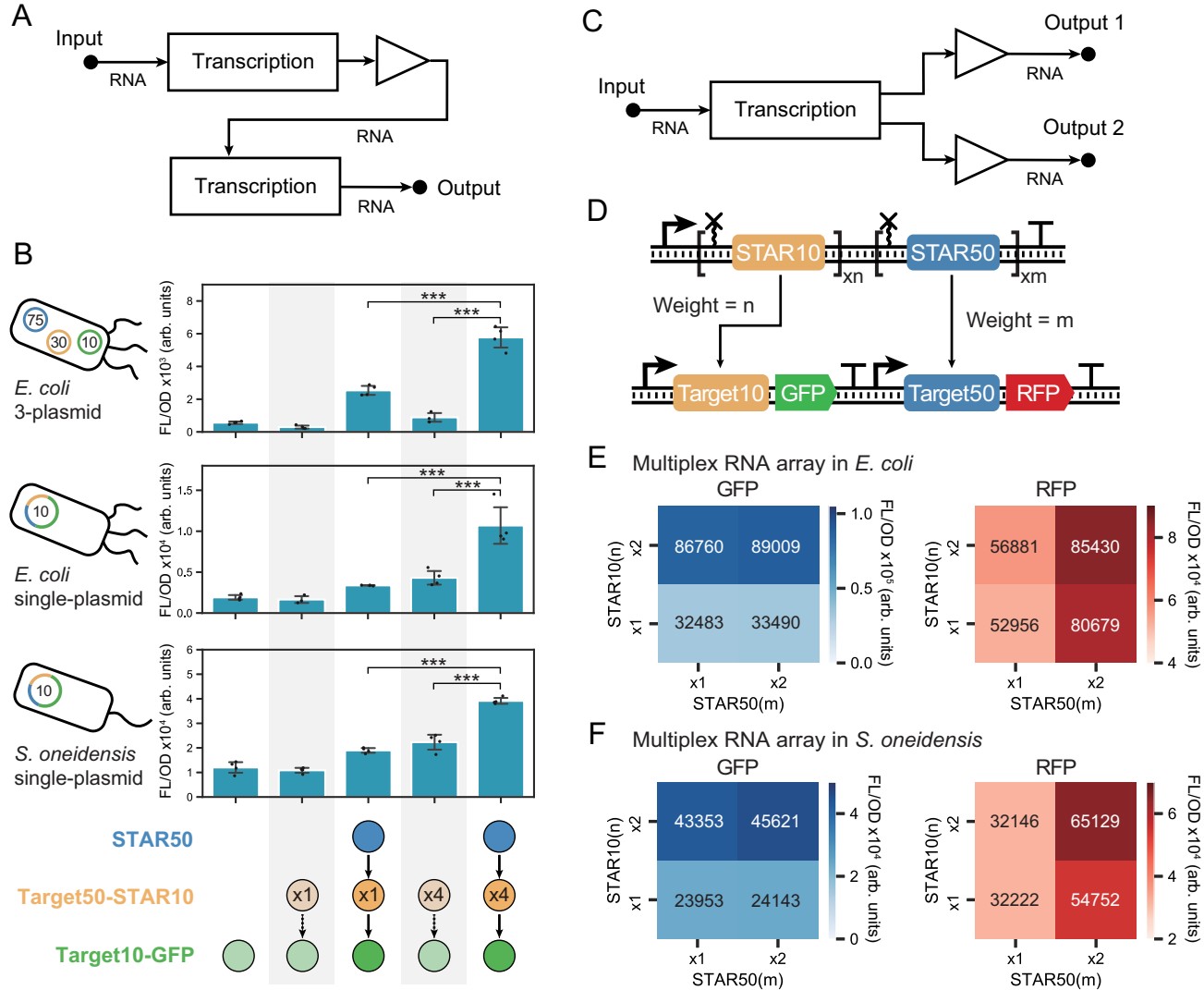

**Fig. 4 | Regulatory RNA arrays allow for tunable and portable RNA-only circuits. A** Schematic of RNA-only activation-activation cascade. The regulatory RNA array functions as an amplifier between two transcription nodes of the RNA cascade. **B** Regulatory RNA arrays serve as signal amplifiers in RNA cascades. Fluorescent characterization in cells transformed with an RNA activation-activation cascade in which STAR50 activates the production of STAR10, which in turn activates a GFP output. From top to bottom panel are a 3-plasmid cascade in *E. coli*, a single plasmid cascade in *E. coli*, and a single plasmid cascade in *S. oneidensis*. Cartoons on left illustrate the plasmid number and expected copy (number in the center of plasmid) with colors corresponding to the network schematic. Bars show mean values and error bars represent s.d. of $n = 4$ biological replicates shown as points. Two-way ANOVA (STAR50 +/- and STAR10x1/x4 as two categorical variables) with post-hoc Tukey test were used ($df = 6$) and the significance between groups are marked by asterisks indicating $p < .05$ (*), $p < .01$ (**), $p < .001$ (***). Fold

activation for the x1 and x4 regulatory array conditions are: 8.6 and 6.5 fold in 3-plasmid *E. coli*, 2.0 and 2.5 fold in 1-plasmid *E. coli*, and 1.7 and 1.8 fold in *S. oneidensis*. **C** Schematic of multiplex RNA array motif. A single transcription event of a multiplex RNA array can provide different RNA outputs that can be independently tuned. **D** Schematic of genetic constructs encoding a multiplex RNA array. n-copy of STAR10 and m-copy of STAR50 are assembled as a single RNA array, which activate the expression of GFP and RFP. **E** Fluorescent characterization of the multiplex RNA array in *E. coli* and (**F**) *S. oneidensis*. Cells were transformed with broad-host range plasmids containing multiplex RNA arrays with the combinations of 1x or 2x STAR10 and STAR50 copies. GFP activated by STAR10 is marked by blue shades and RFP activated by STAR50 is marked by red shades. The numbers indicate the corresponding mean value of the fluorescent/OD. Source data are provided as a Source Data file.

demonstrate that regulatory RNA arrays could be used to tune the output from an activation-activation signaling cascade. In this circuit, the transcriptional output from one STAR regulator is used to drive the transcription of another orthogonal STAR (Supplementary Fig. 12). While STAR signaling cascades have been described[29], a challenge in maintaining sustainable cascades is the signal attenuation at each layer that causes gradual degradation of circuit function with increased layers, a problem that is broadly observed in different types of signaling cascades[38,42–45]. RNA arrays provide a potentially simple solution

to this problem. By implementing the regulatory RNA array as an amplifier between each layer in an RNA cascade, the output of each layer can be increased to reduce signal attenuation (Fig. 4A). We first tested this hypothesis using an activation-activation cascade encoded across three plasmids that decrease in copy number along the cascade from high to low. This design strategy has been used previously to ensure sufficient concentrations of regulators are produced from each layer to ensure activation of the subsequent layer[29,40,41]. In this design, an RNA array composed of either 1 (x1) or 4 (x4) copies was used as the

connecting layer. From this, we observed that both cascades were functional and that the cascade with the x4 RNA array achieved significant amplification of the output signal (Fig. 4B). Interestingly the x4 RNA array also amplifies the signal of the cascade in the off state, resulting in overall comparable dynamic ranges for both the x1 and x4 copy designs (8.6 and 6.5 fold activation). Importantly, the 4-copy RNA array achieves our objective of mitigating signal attenuation.

We next tested if we could use the regulatory RNA arrays to implement a single plasmid activation-activation cascade design. As expected, a drop in dynamic range is observed in the single-plasmid cascade for both the x1 and x4 copy designs (2.0 and 2.5 fold activation); however, using a x4 RNA array reduces signal attenuation resulting in an increase in the overall output signal (Fig. 4B). Given that the single plasmid cascade greatly simplifies the deployment of this system across species, we next tested to see if this cascade would function in *S. oneidensis*. As observed in *E. coli*, we see in *S. oneidensis* that the cascade is functional and that the output signal can be amplified by an RNA array.

We next wanted to explore the applications of regulatory RNA arrays in other types of circuit design. Based on the linear activation observed in Fig. 3, we posited it was possible to create an RNA array that implemented a tunable multiplex network motif in which each output could be independently and precisely weighted, which resembles a fundamental structure in artificial neural networks (ANN) (Fig. 4C). Additionally, this design would be expected to be robust since the multiplex RNA outputs are transcribed as a single transcript through the same promoter thus the output ratios are strictly defined by the copy numbers of STARs. To test this, we aimed to design a series of 1-to-2 multiplex network motifs that used STAR10 and STAR50 with independently control weights (i.e., copies). Specifically, we constructed four variations of this motif where the weight of STAR10 and STAR50, referred to as *n* and *m*, were set to either 1 or 2 copies (Fig. 4D). Output from these arrays was characterized using fluorescence measurements using STAR10 and STAR50 to control a GFP and RFP reporter respectively in *E. coli* cells. As expected, regardless of the exact motif composition, increasing the copy number of STAR10 leads to higher GFP, while increased STAR50 leads to higher RFP demonstrating that this design achieves independent and predictable tuning (Fig. 4E) (Supplementary Fig. 13). Inspired by these results from *E. coli*, we also tested these designs in *S. oneidensis* and observed remarkably similar results (Fig. 4F) (Supplementary Fig. 13). Taken together, these results show that regulatory RNA arrays are composable and tunable for broad-host genetic circuitry.

## Discussion

We have reported the development of a portable regulatory RNA array design motif that allows for tunable and complex regulation across different bacteria. Inspired by natural CRISPR arrays and their processing mechanisms, RNA arrays offer a modular architecture to tune the gain from 1 to 8-fold by changing STAR copy number. This elegant solution to gain tuning relies upon the precise stoichiometric operation of STARs, which we anticipate could be applied to other RNA-based systems. We demonstrated that RNA arrays function across different species and enable the implementation of RNA circuitry that includes activation-activation cascades and tunable multiplex motifs.

A main feature of our study was to demonstrate that STARs have the potential to provide a broad-host tool for gene activation in diverse cellular contexts. This is important because while a versatile set of tools are available for model species like *E. coli*, endeavors to genetically engineer other species can be hindered by a lack of genetic control elements. With a minimal set of genetic elements, STARs can be used to turn on gene expression, fine-tune expression levels, and enact genetic circuitry. We note that a limitation of this study is that we only considered genetically tractable Gram-negative species and future studies are required to understand the host range in Gram-

positive microbes or even more distant cellular contexts such as eukaryotes. Additionally, to implement STARs in genetically intractable microorganisms other innovations will be required, for example, identification of suitable DNA plasmids[46]. Given STARs are functional with bacteriophage RNAP there is also an exciting potential to couple them with the Universal Bacterial Expression Resource (UBER) system[47]. UBER combines a minimal set of broad-host parts and feedback control loops for T7 RNAP to implement an orthogonal transcriptional system that is portable across diverse species.

Our RNA design motif provides a simple approach for tuning output gain that we anticipate will advance RNA synthetic circuitry and applications. For example, regulatory RNA arrays can be used to amplify signals in RNA cascades to counteract signal attenuation and allow for the creation of longer cascades or single-copy circuits which have been achieved using protein transcription factors[48] and CRISPR-Cas regulators[10]. While we have focused on activating RNA in this work, we anticipate the same strategy can be applied to create RNA repressors arrays composed of transcriptional attenuators[40] or STAR sequesters that repress transcription[38,49]. Additionally, regulatory RNA arrays provide a modular approach to tune the weights of transcriptional outputs within RNA network motifs. This characteristic makes them an ideal platform to construct complex circuits including ANN. Compared to protein-based transcription factors that have achieved weight tuning through the creation and screening of promoter mutant libraries[50], regulatory RNA arrays could provide a more direct and predictable means to achieve tunability, which facilitates the optimization of weights in ANN through backpropagation. In the future, we envisage RNA arrays can be used to create RNA-based ANN in cells that can perform complex computations and make precise decisions that are useful for biomedical applications, for example, in the context of classification of healthy/diseased states and programmed drug secretion[51].

In summary, our study contributes a route toward the systematic design of RNA circuitry, adding to other elegant approaches that are uniquely tailored to RNA[9,52,53]. In particular, our results will make it possible to expand RNA synthetic biology to many new exciting applications in diverse, non-model cellular systems.

## Methods

### Plasmid assembly

The information of all plasmids used in this study can be found in Supplementary Table 1. The key plasmid sequences are visualized in Supplementary Figure 14 and an example is provided in Supplementary Table 2. The SP6 promoter sequences are provided in Supplementary Table 3. The regulatory RNA array cassettes were created using a modular cloning strategy shown in Supplementary Figure 2. The examples of primer and overhang sequences used for the modular cloning approach are shown in Supplementary Table 4. The broad-range plasmids containing regulatory RNA arrays were further constructed through Gibson assembly. All assembled plasmids were verified using Sanger DNA sequencing. Plasmid maps are provided in the Supplementary Data 1 file.

### Plasmid transformation

For experiments using *E. coli* cells, plasmids were transformed into chemically competent *E. coli* (TG1 strain). *E. coli* cells were then plated on LB-agar plates containing combinations of 100 µg/mL carbenicillin (Sigma-Aldrich), 34 µg/mL chloramphenicol (Sigma-Aldrich), 50 µg/mL spectinomycin (Sigma-Aldrich), 100 µg/mL kanamycin (Sigma-Aldrich) depending on the plasmids used, and incubated overnight at 37 °C. For experiments using *S. oneidensis*, *P. fluorescens*, and *P. putida*, plasmids were transferred into electrocompetent cells. These cells were prepared by washing overnight-cultured cells 3 times with 10% glycerol. Plasmids were electroporated into these cells using a MicroPulser Electroporator (Bio-rad) with 1.2 kV, 1.25 kV, or 2.5 kV

pulse. The cells were then recovered at 30 °C in LB without antibiotics for 2 hours and plated on LB plates with 50 μg/mL kanamycin and incubated overnight at 30 °C. For experiments using *P. stutzeri*, the overnight-cultured cells were washed 3 times with 300 mM sucrose and then plasmids were electroporated using a 2.5 kV pulse. The cells were recovered at 30 °C in LB for 2 hours and plated on LB plates with 50 μg/mL kanamycin and incubated overnight at 30 °C. For experiments using *V. natriegens*, plasmids were electroporated into electrocompetent cells made from *V. natriegens* cells mixed in electroporation buffer (680 mM sucrose, 7 mM K2HPO4, pH 7) and pulsed at 0.7 kV. Electroporated cells were recovered at 37 °C for 1 hour, plated onto LB3-agar (25 g/L LB broth [Fisher], 20 g/L NaCl [Fisher], 15 g/L Agar [Fisher]) with 200 μg/mL kanamycin, and incubated overnight at 37 °C. Relevant conditions are also included in Supplementary Table 5.

### Culturing conditions
For experimental measurements, four colonies were used to inoculate separate liquid cultures of 200 μL of LB (LB3 for *V. natriegens*) containing corresponding antibiotic in a 2 mL 96-well block (Costar), and grown overnight (-18 hours) at 37 °C for *E. coli* and *V. natriegens* or 30 °C for *S. oneidensis*, *P. fluorescens*, *P. putida*, and *P. stutzeri*. All cultures were grown at 1000 rpm in a VorTemp 56 bench top shaker (Labnet). For experiments without inducible promoters, 4 μL of each overnight culture were added to 196 μL (1:50 dilution) of LB (LB3 for *V. natriegens*) containing the corresponding antibiotic in a newly prepared 96-well block. The new block was incubated at 1000 rpm for 8 hours at 37 °C (*E. coli* and *V. natriegens*), 6 hours at 30 °C (*S. oneidensis* and *P. fluorescens*), and 12 hours at 30 °C (*P. putida* and *P. stutzeri*). For experiments with arabinose-inducible promoters, 4 μL of each overnight culture were added to 192 μL of LB containing corresponding antibiotics in a newly prepared 96-well block. After 4 hours of incubation, 4 μL of 0.5%, 1.5%, 5%, and 15% arabinose solution was added to reach a final concentration of 0.01%, 0.03%, 0.1%, and 0.3% respectively, and grown for another 4 hours before the measurement. For experiments with cumate-inducible plasmids, 4 μL of overnight culture were added to 196 μL of LB containing corresponding antibiotics and 10 μM, 30 μM, 60 μM, or 100 μM cumate, and grown for 8 hours before the measurement. Relevant conditions are also included in Supplementary Table 5.

### Fluorescence measurement
Bulk fluorescence measurements were performed with 50 μL of experimental culture diluted in 50 μL of phosphate buffered saline (PBS, (Fisher)) in a 96-well plate. The GFP was measured with 485/520 mm as excitation/emission wavelength. The RFP was measured with 560/630 mm as excitation/emission wavelength. Optical density at 600 nm [OD] was also measured.

### Bulk fluorescence data analysis
Each 96-well block included two sets of controls; a media blank and *E. coli* transformed with combinations of empty control plasmids pJEC101, pJEC102, pJEC103, or *S. oneidensis*, *P. fluorescens*, *P. putida*, *P. stutzeri*, and *V. natriegens* without plasmids, referred to here as blank cells. Blank cells were used to determine autofluorescence levels. OD and FL values for each colony were first corrected by subtracting the mean value of the media blank from the respective value of the experimental conditions. The ratio of the corrected FL to the corrected OD (FL/OD) was then calculated for each well. Autofluorescence was removed through the subtraction of FL/OD of the blank cells.

### RNA secondary structure prediction
The online version of Nucleic Acids Package (NUPACK) version 4.0.0.27 was used to predict the secondary structure of RNA species. The insulator sequence and an example of regulatory RNA array

sequence used in NUPACK simulation are shown in Supplementary Table 6.

### Statistical analysis
The mean and standard deviation were calculated from the biological replicates of each experiment. Two-tailed t-tests assuming unequal variance were used to calculate p-values. The statistical significance was marked with $p < .05$ (*), $p < .01$ (**), $p < .001$ (***). The Pearson correlation coefficient was calculated between the copy number of the RNA array and fluorescence/OD to evaluate their linear correlation. One-way and two-way ANOVA with post-hoc Tukey test were used and the significance between groups are marked by asterisks indicating $p < .05$ (*), $p < .01$ (**), $p < .001$ (***).

### Model simulations
All models (reported in Supplementary Note 1) were computationally simulated by solving ODEs using Python over a set of discrete time steps using estimated parameters.

### Reporting summary
Further information on research design is available in the Nature Portfolio Reporting Summary linked to this article.

## Data availability
The data generated in this study are provided in the Source Data file. Source data are provided with this paper.

## Code availability
Source code is provided with GitHub link: github.com/ccubasam-code/RNA-based-circuits[54].

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

## Acknowledgements

The authors acknowledge the Chappell Lab members for helpful discussion and Shyam Bhakta for input on the modular cloning approach. This material is based on work supported by the National Science

Foundation award no. 2124306 (J.C.) and BBSRC/BIO award no. 2020039 (E.F.) and the National Institutes of Health through the joint NSF-National Institutes of General Medical Sciences Mathematical Biology Program grant no. RO1GM144959 (MRB). J.C. is an Alfred P. Sloan Research Fellow.

## Author contributions

B.L., C.C.S., M.B., E.F., and J.C. contributed to the design of experiments, collecting of data, and writing of the manuscript.

## Competing interests

The authors declare no competing interests.
