## [Peer Review File · Nature Communications]

A portable regulatory RNA array design enables tunable and complex regulation across diverse bacteriaREVIEWER COMMENTS

Reviewer #2 (Remarks to the Author):

Liu et al describe the use of small transcriptional activating RNA (STAR) to control gene expression in a range of Gram-negative species by using bacterial and phage RNA polymerases. They then go on to tune the output of GFP by introducing varying numbers of copies of STAR to further amplify the fluorescent signal, which is then extended to include RFP in a multiplex array format. This work focusses on tuning GFP and RFP, and would be improved by tuning the output of different, non-fluorescent proteins that can better show the future application of the work. I am also curious about the stability of the strains with multiple copies of STAR, could the authors provide some comment or suggestion with this regard? Generally, the publication is interesting and therefore I suggest that it is suitable for publication in Nature Communications.

Specific comments:

Figure 1 part D, showing transcription by SP6 phage polymerase could be made simpler to read by including the fact that this is expressed in *E. coli* in the figure itself.

Figure 3, part F is difficult to follow. Please provide further explanation to aid understanding. For example, details should be given on what the units for transcription input and output RNA represent. Does a transcription input of 1 indicate that all copies of this gene have been transcribed? Are the units of output RNA arbitrary?

Full sequence information for all plasmids should be provided in the supplementary data to allow the work to be repeated.

Reviewer #3 (Remarks to the Author):

The authors build upon their previous work of small transcription activating RNAs (STARs) to activate transcription. In this new manuscript they first show that STARs also work in other gram-negative bacteria than *E. coli* and with the phage SP6RNAP instead of the bacterial RNAP. Next, they show that multiple STARs can increase the activation. Finding the right way of insulating the different copies of the STARs was crucial. Then, they show that the relation of number of STARs and the activation strength is linear (at least under certain conditions) in *E. coli* and *S. oneidensis*. Finally, they used this setup to build a simple cascade as well as a circuit that controls two outputs (both in *E. coli* and *S. oneidensis*).

The manuscript is well written and the figures are clear. The approach has its value, especially as it seems transferable between bacteria. The methods are detailed enough.

Major comments:

- The introduction talks a lot about non-model organisms. However, for example *Pseudomonas putida* is studied in many labs, used in industrial applications and we have already a nice genetic toolbox for this bacterium. Also the other bacteria used in this work, have been already engineered. It is nice that the STARS are transferable between a selection of gram-negative bacteria. However, this work does not really address the problem of non-model organisms, where common plasmids might not replicate and we do not know promoters that work. I suggest, you tone down your claims about non-model organisms in the introduction and/or discuss the challenges of synthetic biology in “real” non-model organisms.
- Fig. 3B-E: You have linearity at high concentrations of your inducers. Please also show what happens at lower concentrations (e.g. in a supporting figure). I suspect that it is not linear. If not, please discuss the limitations of the linearity and adjust your claims about linearity.
- Fig. 3F: Similar to the experimental data, you only assess linearity at one point (0.5 transcriptional input). At higher values, it would clearly not be linear anymore. Please discuss those limitations. The model only shows up to 4 copies, while in the experiment you used up to 8 copies. Please include also up to 8 copies in your model.
- Fig. 4B: It would be important to show here the case of 0 inducer of the whole system to assess the leakiness. This seems more relevant than showing partial circuits as shown now.
- I could not find the source data, even though it says in the text it is provided

Minor comments:

- Fig 4, Fig S3: You compare here more than 2 groups with each other. In this case, t-tests don't seem appropriate, but a non-parametric ANOVA test would be required
- Fig. 3: please also provide p values for your correlation coefficients (r)
- Fig. 3: also add values at 0 inducer concentration.

- Fig S6 and Fig.3: I don't understand why you do this data conversion from inducer concentration to promoter activity. Please either explain what the reason is or show directly data of Fig. S6 in Fig. 3.
- You mostly show data of STAR10 and STAR50 and often they differ quite a bit in their fold-induction. Have you tested other STARs from your library, and how much is the variation from STAR to STAR?
- Fig. 1C/D, Fig. 2B (middle panel), Fig 4B: It would be interesting to see how much of the condition without STAR differs from to the autofluorescence of the bacteria. From the methods I understand that you have also measured that. I suggest that you subtract the autofluorescence from the values. Also, it would be nice to have a number for the fold-improvement. I suggest you write this number above the bar plots.
- Fig. 2B middle: It would be interesting to know how much the activation goes down by attaching an insulator. Could you compare a 1x STAR with and without insulator (including processing) (e.g. as supporting figure)?
- Fig 2B bottom/Supporting Fig. 4: how do the predicted structures look like after processing? How does the structure of a single STAR without insulator look like?
- Fig. 4B: what would be the induction of GFP if the input would directly induce STAR10 in one or 4 copies? These data are in Fig. 2, but it would be nice to have it here in the same plot.
- Fig. 4B: Why does your X4-STAR10 construct show already a tendance to higher fluorecence even in the absence of the previous layer? Is this due to the leakiness?
- Fig. 4E (+Fig. S8): It is difficult to guess the actual values from the colors in the heatmap. Please write the actual values into the squares. Also, please also comment what might be the reason for the RFP changes depending on the number of STAR10 copies.
- Fig. 1D is not referenced in the text
- Supplementary Fig. 8 is not referenced in text

- “In most cases we do not appear to observe saturation” should be “In most cases we do not observe saturation”
- It is good that you provide a DNA plasmid sequence. However, there are better ways of sharing DNA sequences than as text in a pdf. I suggest for example an annotated gb file. You can then directly provide all the plasmid sequences, not just one example.
- “Interesting, as shown for STARS” should be “Interestingly, as shown for STARS” Supplementary Table 1 is very good. However, the figure legends should still also mention which plasmids were used for the experiments shown
- STAR can only activate transcription. However, for complex circuits, a combination of activation and repression is necessary. Can you add a short discussion about how you would combine your approach with an approach that can provide repression?
- Other labs would adapt your strategy more easily, if you would provide your plasmids on Addgene.

Reviewer #2 (Remarks to the Author):

Liu et al describe the use of small transcriptional activating RNA (STAR) to control gene expression in a range of Gram-negative species by using bacterial and phage RNA polymerases. They then go on to tune the output of GFP by introducing varying numbers of copies of STAR to further amplify the fluorescent signal, which is then extended to include RFP in a multiplex array format. This work focusses on tuning GFP and RFP, and would be improved by tuning the output of different, non-fluorescent proteins that can better show the future application of the work. I am also curious about the stability of the strains with multiple copies of STAR, could the authors provide some comment or suggestion with this regard? Generally, the publication is interesting and therefore I suggest that it is suitable for publication in Nature Communications.

Comments and Response:

(1) In response to “I am also curious about the stability of the strains with multiple copies of STAR, could the authors provide some comment or suggestion with this regard?” mentioned in the summary paragraph:

We thank the reviewer for this insightful comment and agree that genetic stability is important to investigate. To do this, we have now included a consecutive culturing experiment in which cultures of cells containing the RNA arrays were grown and diluted over 5 days, plasmids isolated, and sequencing performed. For this experiment, we focused on the 8-copy RNA array designs, which we reasoned would be most affected by genetic instability. We observed no significant recombination or mutations in these plasmids after the 5-day culture, suggesting high stability of the RNA array plasmids. We did observe a small dimer peak in all sequencing results including the non-array control plasmids (pBL301, pBL303, pBL401), which we reason is due to the natural multimerization of pBBR1 plasmids that we and others have observed. We include this new supplementary figure 10 below:

Supplementary Fig. 10. Genetic stability test of regulatory RNA array. Plasmids with pBAD-STAR10x8 and pCymR-STAR10x8 were transformed into *E. coli* (TG1) and cultured overnight on LB plate with kanamycin. Single colonies were picked and inoculated in the culture tubes containing 5 mL LB with kanamycin at 37 °C. The bacteria were successively inoculated (1:1000 dilution) and cultured over 5 days. The plasmids were extracted through mini-prep and sent for full plasmid sequencing (plasmidsaurus) on day 1, 3, and 5. The size distributions of plasmid sequencing reads are shown in the bar plots. This analysis showed comparable sequencing results across the days tested and no mutations in RNA arrays were observed. The sequencing data is included in the supplementary files.

(2) Figure 1 part D, showing transcription by SP6 phage polymerase could be made simpler to read by including the fact that this is expressed in *E. coli* in the figure itself.

We thank the reviewer for this comment and have subsequently adjusted the figure to more clearly state that these experiments were performed in *E. coli* cells using expressed SP6 RNAP.

(3) Figure 3, part F is difficult to follow. Please provide further explanation to aid understanding. For example, details should be given on what the units for transcription input and output RNA represent. Does a transcription input of 1 indicate that all copies of this gene have been transcribed? Are the units of output RNA arbitrary?

We thank the reviewer for this comment to improve the interpretation of the model. The figure legend has been adjusted to clarify the points raised. Specifically, the transcription input is the transcription rate of the promoter driving the array's transcription and the output RNA is the concentration of the RNA product produced by the target RNA. We also note that the units used are arbitrary units.

- (4) Full sequence information for all plasmids should be provided in the supplementary data to allow the work to be repeated.**

We thank the reviewer for this comment and have uploaded the plasmid sequence files. We additionally intend to upload key plasmids to Addgene.

Reviewer #3 (Remarks to the Author):

The authors build upon their previous work of small transcription activating RNAs (STARs) to activate transcription. In this new manuscript they first show that STARs also work in other gram-negative bacteria than *E. coli* and with the phage SP6RNAP instead of the bacterial RNAP. Next, they show that multiple STARs can increase the activation. Finding the right way of insulating the different copies of the STARs was crucial. Then, they show that the relation of number of STARs and the activation strength is linear (at least under certain conditions) in *E. coli* and *S. oneidensis*. Finally, they used this setup to build a simple cascade as well as a circuit that controls two outputs (both in *E. coli* and *S. oneidensis*).

The manuscript is well written and the figures are clear. The approach has its value, especially as it seems transferable between bacteria. The methods are detailed enough.

Major comments:

- (1) The introduction talks a lot about non-model organisms. However, for example *Pseudomonas putida* is studied in many labs, used in industrial applications and we have already a nice genetic toolbox for this bacterium. Also the other bacteria used in this work, have been already engineered. It is nice that the STARs are transferable between a selection of gram-negative bacteria. However, this work does not really address the problem of non-model organisms, where common plasmids might not replicate and we do not know promoters that work. I suggest, you tone down your claims about non-model organisms in the introduction and/or discuss the challenges of synthetic biology in “real” non-model organisms.**

We thank the reviewer for this insightful comment. We agree that the diverse Gram-negative species we used have some established genetic parts and a history of being engineered, albeit significantly less than *E. coli*. We have carefully checked our claims to ensure we have not claimed to engineer non-model species. As we do believe our work is an important step toward engineering non-model species (*i.e.*, through the creation of portable RNA parts), we have kept the discussion of non-model microbes in the introduction, which was a primary motivation for this work. However, we have included an expanded discussion of the existing challenges that need to be overcome to achieve the reliable engineering of non-model species, such as the discovery of plasmids that can be introduced into these species.

- (2) Fig. 3B-E: You have linearity at high concentrations of your inducers. Please also show what happens at lower concentrations (e.g. in a supporting figure). I suspect**

that it is not linear. If not, please discuss the limitations of the linearity and adjust your claims about linearity.

We thank the reviewer for this interesting question and suggested analysis. We agree that this is important to show and have now included a new Supplementary Figure 8 that shows the relation between STAR copy and transcription output at different induction levels for the two different promoters used that represent a low and medium level of induction. Interestingly, this analysis showed similar linearity in other induction levels, suggesting robust amplification across a range of transcription inputs. We include this new figure and legend below:

Supplementary Fig. 8. Relation between STAR copy and fluorescent output across different transcription input levels. The correlation between the STAR copy number in the RNA array and the corresponding fluorescent output with different inducer concentrations. Graphs show data from (A) pBAD in *E. coli*, (B) pCymR in *E. coli*, (C) pBAD in *S. oneidensis*, and (D) pCmyR in *P. fluorescens*. Pearson correlation coefficient and p-value is calculated and shown as r and p .

(3) Fig. 3F: Similar to the experimental data, you only assess linearity at one point (0.5 transcriptional input). At higher values, it would clearly not be linear anymore. Please discuss those limitations.

We agree with the reviewer that with a high transcriptional input the model does not hold a linear relationship between copy number and output. When the concentration of RNA input is higher than the number of inactive complexes, the latter becomes the limiting factor in the output. As a result, we observe a saturation of the output for larger transcriptional input. The simple model suggests that the higher copies result in reaching the saturation regime for lower transcriptional

input. Interestingly, in most cases we do not observe saturation in our experimental data, suggesting we are below this threshold. The one exception to this is in *E. coli* using a pBad input (Figure 3B), which appears to begin to saturate at high induction levels.

To address this, we have also included an additional supplementary figure, in which we show the relation between input and output at different induction levels (*e.g.*, saturating and non-saturating conditions). This new Supplementary Figure 11 is shown below in response to the next comment. Here, in the right panel we show the linearity does indeed break down at high levels of transcription input.

(4) The model only shows up to 4 copies, while in the experiment you used up to 8 copies. Please include also up to 8 copies in your model.

We thank the reviewer for this comment. To address this, we incorporated a new figure in the supplementary information of the simulations for up to 8 STAR copies. We also simulate for different transcriptional inputs below the saturation regime (linear relationship), and close to the saturation regime (loss of linear relationship). This new figure is pasted below.

Supplementary Fig. 11. Model predictions of regulatory RNA arrays. (Left panel) The input-output behavior between transcription input and output RNA for multiple STAR from 1 to 8 copies. The dotted line indicates full cleavage and the solid line indicates partial cleavage of the RNA array. (Right panel) The relationship between number of repeats and output for three different transcriptional inputs from $\theta = 0.1$ to 0.8.

(5) Fig. 4B: It would be important to show here the case of 0 inducer of the whole system to assess the leakiness. This seems more relevant than showing partial circuits as shown now.

We thank the reviewer for this insightful suggestion. In this case the cascade is under the control of a constitutive promoter, meaning it is not possible to show the 0 inducer condition. However, we do agree that the leakiness of the cascade is important to convey and believe that our partial

circuit characterizations convey this. For example, by including the bottom two nodes of the cascade in the absence of the top node, we are in essence showing how much leaky expression occurs due to the inherent leak of the Target-STAR construct when transcribed with a strong constitutive promoter. We believe that this (and the other conditions) will allow readers to understand the leakiness in the cascade.

(6) I could not find the source data, even though it says in the text it is provided

We thank the reviewer for this observation and have now included the source data.

Minor comments:

(7) Fig 4, Fig S3: You compare here more than 2 groups with each other. In this case, t-tests don't seem appropriate, but a non-parametric ANOVA test would be required

We thank the reviewer for this suggestion and have now included new statistical analysis with ANOVA tests.

(8) Fig. 3: please also provide p values for your correlation coefficients

We thank the reviewer for this comment and have now included these.

(9) Fig. 3: also add values at 0 inducer concentration.

We thank the reviewer for this comment and have now included this as a new Supplementary Figure 9 and discussion in our results section. Interestingly, the STAR copy did not appear to significantly alter the off-state expression in the absence of the inducer (*i.e.*, leakiness). This new figure and legend below:

Supplementary Fig. 9. Characterization of leakiness in regulatory RNA arrays using different STAR copies. (A) The full set of inducible regulatory RNA arrays were tested without induction together with no target-GFP control, no STAR control, and constitutive STAR control in *E. coli*, *S. oneidensis*, and *P. fluorescens*. (B) A parallel test was done with 0.3% arabinose or 100 μ M cumate as positive control.

(10) Fig S6 and Fig.3: I don't understand why you do this data conversion from inducer concentration to promoter activity. Please either explain what the reason is or show directly data of Fig. S6 in Fig. 3.

We thank the reviewer for this comment. Our goal with converting the data from inducer concentration to promoter activity was because the relationship between the inducer concentration and promoter activity is often non-linear. In this case, this could complicate the interpretation of our amplification data and as such, sought to present the effect of our amplifiers on the relationship between transcription input and output. As noted, we have been careful to include all data. We have adjusted the text to make this clear, which is pasted below for your convenience.

“As the relation between inducer concentration and the relative promoter activity (*i.e.*, transcription output) of different inducible systems is often non-linear, we converted the inducer concentrations into a relative promoter activity term. To do this, we measured the transcription output of each inducible promoter system using a GFP reporter under the same experimental conditions to create a series of standard curves (Supplementary Fig. 6). This was used to convert the inducer concentration used in the array characterization into an expected input promoter activity (Supplementary Fig. 7).”

(11) You mostly show data of STAR10 and STAR50 and often they differ quite a bit in their fold-induction. Have you tested other STARs from your library, and how much is the variation from STAR to STAR?

We thank the reviewer for this interesting question. Our choice of STAR10 and STAR50 was because these STAR variants were part of an orthogonal STAR library previously reported (<https://doi.org/10.1038/s41467-017-01082-6>). We have focused on testing these two orthogonal variants to allow us to construct simple genetic circuits (e.g., cascades). As noted, between each STAR variant the off state, on state, and overall fold activation can vary, which is reported in the original manuscript (<https://doi.org/10.1038/s41467-017-01082-6>). While we have only tested these STARs, we do not anticipate challenges in applying our strategy to other STAR variants. In rare cases, there could be issues with our approach if the STAR forms an interaction with the insulators that results in misfolding of these structures.

(12) Fig. 1C/D, Fig. 2B (middle panel), Fig 4B: It would be interesting to see how much of the condition without STAR differs from to the autofluorescence of the bacteria. From the methods I understand that you have also measured that. I suggest that you subtract the autofluorescence from the values. Also, it would be

nice to have a number for the fold-improvement. I suggest you write this number above the bar plots.

We thank the reviewer for this comment. In the data presented, we have corrected for autofluorescence and have adjusted the methods to make this more clear. Additionally, we have reported the fold activation for the 1C/D and Fig 4B in the figure legend.

- (13) **Fig. 2B middle: It would be interesting to know how much the activation goes down by attaching an insulator. Could you compare a 1x STAR with and without insulator (including processing) (e.g. as supporting figure)?**

We thank the reviewer for this comment and have included a new Supplementary Fig 5 and discussion to address this comment. From this we confirmed that a single STAR maintained similar performance after the addition of *csy4hp* or *shcsy4hp*, suggesting robust folding of these insulators. This new figure is copied below:

Supplementary Fig. 5. Impact of insulators on STAR activation. Fluorescent characterization of a 3-plasmid system containing STAR, Target-GFP, and Csy4 endonuclease. From left to right: STAR10 without insulator, no STAR, STAR10 with *csy4hp*, STAR10 with *shcsy4hp*, and STAR10 with *shcsy4hp* but without Csy4 endonuclease. Bars show mean values and error bars represent s.d. of $n = 4$ biological replicates shown as points.

- (14) **Fig 2B bottom/Supporting Fig. 4: how do the predicted structures look like after processing? How does the structure of a single STAR without insulator look like?**

We thank the reviewer for this interesting question. To answer this, we show below the predicted minimum free energy (MFE) structures of the STAR10 with and without a processed insulator. As

you can see, the insulation only amends a small 'unstructured' RNA sequence onto the STAR, minimizing its impact on the overall STAR fold.

STAR10 with insulator after processing

- (15) **Fig. 4B: what would be the induction of GFP if the input would directly induce STAR10 in one or 4 copies? These data are in Fig. 2, but it would be nice to have it here in the same plot.**

We thank the reviewer for this suggestion. In this case, we decided not to add this data to Figure 4B to avoid increasing the density of this figure.

- (16) **Fig. 4B: Why does your X4-STAR10 construct show already a tendence to higher fluorescence even in the absence of the previous layer? Is this due to the leakiness?**

This is a great observation. As the reviewer noted, the increased background is indeed due to the leakiness of the STAR. We have adjusted the manuscript to note this.

- (17) Fig. 4E (+Fig. S8): It is difficult to guess the actual values from the colors in the heatmap. Please write the actual values into the squares. Also, please also comment what might be the reason for the RFP changes depending on the number of STAR10 copies.**

We thank the reviewer for this useful suggestion and have the numbers shown in the heatmap. The higher RFP in the presence of 2xSTAR10 is probably due to the transcriptional leakiness from the upstream promoter activated by STAR10.

- (18) Fig. 1D is not referenced in the text**

We thank the reviewer for highlighting this and have now amended the text.

- (19) Supplementary Fig. 8 is not referenced in text**

We thank the reviewer for highlighting this and have now amended the text.

- (20) “In most cases we do not appear to observe saturation” should be “In most cases we do not observe saturation”**

We thank the reviewer for highlighting this and have now amended the text.

- (21) It is good that you provide a DNA plasmid sequence. However, there are better ways of sharing DNA sequences than as text in a pdf. I suggest for example an annotated gb file. You can then directly provide all the plasmid sequences, not just one example.**

We thank the reviewer for this valuable suggestion and have now included annotated gb files with our submission.

- (22) “Interesting, as shown for STARS” should be “Interestingly, as shown for STARS**

We thank the reviewer for highlighting this and have now amended the text.

- (23) ”Supplementary Table 1 is very good. However, the figure legends should still also mention which plasmids were used for the experiments shown**

We thank the reviewer for this suggestion. In this case, we have decided not to include the plasmid number used for each figure legend as this would increase the length and density of these legends. We believe that our current naming strategy used in the figures and the supplementary table 1 are sufficient for interested readers to identify which plasmids are used.

(24) STAR can only activate transcription. However, for complex circuits, a combination of activation and repression is necessary. Can you add a short discussion about how you would combine your approach with an approach that can provide repression?

We thank the reviewer for this interesting comment. We indeed believe it is possible to use our strategy with RNA repressors and have added an additional discussion of this in our discussion section, which is pasted below for your convenience:

“While we have focused on activating RNA in this work, we anticipate the same strategy can be applied to create RNA repressors arrays composed of transcriptional attenuators or STAR sequesters that repress transcription.”

(25) Other labs would adapt your strategy more easily, if you would provide your plasmids on Addgene.

We thank the reviewer for this comment and intend to deposit key plasmids on Addgene.

REVIEWERS' COMMENTS

Reviewer #2 (Remarks to the Author):

The authors have addressed this reviewers comments

Reviewer #3 (Remarks to the Author):

The authors have addressed all my queries in a satisfactory manner. I support its publication.

Yolanda Schaerli